# Investigating the quality of HIV rapid testing practices in public antenatal health care facilities, South Africa

**Duduzile F. Nsibande**[1,2]*, **Selamawit A. Woldesenbet**[3,4], **Adrian Puren**[3], **Peter Barron**[4], **Vincent I. Maduna**[5], **Carl Lombard**[6,7], **Mireille Cheyip**[8], **Mary Mogashoa**[8], **Yogan Pillay**[9], **Vuyolwethu Magasana**[1,2], **Trisha Ramraj**[1,2], **Tendesayi Kufa**[3,4], **Gurpreet Kindra**[8], **Ameena Goga**[1,2,10☯], **Witness Chirinda**[1☯]

1 Health Systems Research Unit, South African Medical Research Council, Cape Town, South Africa, 2 HIV and other Infectious Diseases Research Unit, South African Medical Research Council, Cape Town, South Africa, 3 Center for HIV and STI, National Institute for Communicable Diseases, Johannesburg, South Africa, 4 School of Public Health, University of the Witwatersrand, Johannesburg, South Africa, 5 Directorate of Research & Innovation, Tshwane University of Technology, Pretoria, South Africa, 6 Biostatistics Unit, South African Medical Research Council, Cape Town, South Africa, 7 Division of Epidemiology and Biostatistics, Department of Global Health, University of Stellenbosch, Cape Town, South Africa, 8 United States Centers for Disease Control and Prevention, Pretoria, South Africa, 9 National Department of Health, Pretoria, South Africa, 10 Department of Paediatrics and Child Health, University of Pretoria, Pretoria, South Africa

☯ These authors contributed equally to this work.
* Duduzile.Nsibande@mrc.ac.za

**Data Availability Statement:** All data files are available from the Figshare data repository: DOI 10.6084/m9.figshare.20257362.

## Abstract

Monitoring HIV prevalence using antenatal HIV sentinel surveillance is important for efficient epidemic tracking, programme planning and resource allocation. HIV sentinel surveillance usually employs unlinked anonymous HIV testing which raises ethical, epidemiological and public health challenges in the current era of universal test and treat. The World Health Organization (WHO) recommends that countries should consider using routine prevention of mother-to-child transmission of HIV (PMTCT) data for surveillance. We audited antenatal care clinics to assess the quality of HIV rapid testing practices as the first step to assess whether South Africa is ready to utilize PMTCT programme data for antenatal HIV surveillance. In 2017, we conducted a cross-sectional survey in 360 randomly sampled antenatal care clinics using the adapted WHO Stepwise-Process-for-Improving-the-Quality-of-HIV-Rapid-Testing (SPI-RT) checklist. We calculated median percentage scores within a domain (domain-specific median score), and across all domains (overall median percentage scores). The latter was used to classify sites according to five implementation levels; (from 0:<40% to 4: 90% or higher). Of 346 (96.1%) facilities assessed, an overall median percentage score of 62.1% (inter-quartile range (IQR): 50.8–71.9%) was obtained. The lowest domain-specific median percentage scores were obtained under training/certification (35% IQR: 10.0–50.0%) and external quality assurance (12.5% IQR: 0.0–50.0%), respectively. The majority (89%) of sites had an overall median score at level 2 or below; of these, 37% required improvement in specific areas and 6.4% in all areas. Facilities in districts implementing the HIV Rapid Test Quality Improvement Initiative and supported by the President's Emergency Plan for AIDS Relief (PEPFAR) had significantly higher median overall scores

**Funding:** This work has been supported in part by the United States President's Emergency Plan for AIDS Relief (PEPFAR) through the US Centers for Disease Control and Prevention (CDC) under the terms of Cooperative Agreement 1U2GGH001150. • Specific grant numbers: Cooperative Agreement 1U2GGH001150 • Initials of authors who received each award: AG • Full names of commercial companies that funded the study or authors: none • Initials of authors who received salary or other funding from commercial companies: none • URLs to sponsors' website: Centers for Disease Control and Prevention (cdc.gov) The funders had no role in study design, data collection and analysis, decision to publish, or preparation of the manuscript.

**Competing interests:** The authors have declared that no competing interests exist.

(65.6% IQR: 53.9–74.2%) (P-value from rank sum test: <0.001) compared with non–PEPFAR–supported facilities (56.6% IQR:47.7–66.0%). We found sub-optimal implementation of HIV rapid testing practices. We recommend the expansion of the PEPFAR-funded Rapid Test Continuous Quality Improvement (RTCQI) support to all antenatal care testing sites.

## Introduction

HIV/AIDS remains a serious public health problem especially in the Sub-Saharan Africa (SSA) which carries 53% of the world's people living with HIV [1]. Periodic monitoring of HIV prevalence is critical for estimating the overall burden of disease, efficient programme planning and resource allocation. For over three decades countries have used traditional unlinked anonymous testing (UAT)-based antenatal survey (ANSUR) methods for monitoring HIV prevalence. Using these however, raise ethical, epidemiological and public health challenges in low income countries [2], especially in the current era of universal test and treat (UTT) strategy as clients cannot be linked to their test results for continuity of care. The World Health Organization (WHO) recommends that countries with near universal prevention of mother-to-child transmission of HIV (PMTCT) coverage migrate from traditional UAT-based ANSUR to routine PMTCT programme-based surveillance methods to monitor trends in HIV prevalence [3]. This transition will facilitate ethically and methodologically sound surveillance as all HIV sero-status data will come from routine HIV testing where pregnant women receive their HIV test result and are linked to HIV treatment or prevention services [2]. This approach will also reduce the workload and financial cost associated with ANSUR, improving sustainability of surveillance and strengthening routine data systems [4].

For routine HIV testing data to be used for surveillance, it must be of good quality and the uptake of routine HIV testing must be above 95% [3]. Inaccuracies and incompleteness of PMTCT facility-based data [5, 6], including misdiagnosis of HIV status have been widely reported [7–10] especially in resource-limited settings. Despite advances in the development of HIV testing technologies, algorithms and policies, routine HIV testing carries a potential of incorrect diagnoses due to technical and human factors that interact at health system, provider, patient and test device levels [11, 12] in most resource-constrained settings [8, 13]. Countries that have explored the utility of routine PMTCT data for surveillance found mixed results [5, 14–18]. Some countries delayed using PMTCT programme-based data for surveillance despite reporting high overall positive percentage agreements (PPAs) in prevalence estimates between ANSUR and routine PMTCT data [5, 17, 18]. This was due to in-country site level differences in HIV prevalence [5, 18], low uptake of PMTCT HIV testing and limited PMTCT data quality [5, 16, 19]. Prior to 2017, South Africa (SA) used UAT-based ANSUR to monitor HIV prevalence in pregnancy and to assist in the overall modelling of HIV prevalence [20].

Evidence suggests that in SA there is substantial sub-population variation in uptake of HIV testing services (HTS) [21], and the 2017 SA ANSUR findings show that over a third (39.2%) of pregnant women visiting antenatal care (ANC) for the first time were not aware of their HIV-positive status [22]. This underscores the need for sustaining universal uptake and good quality antenatal HIV testing. While challenges in the quality of HTS services have been recorded in SA [23, 24], health facility service user satisfaction has been reported to be high (89.8%) [25]. To ensure the fidelity of routine HIV rapid testing, WHO recommends periodic site audits, external quality assurance (EQA) and re-testing of clients to verify HIV status before antiretroviral therapy (ART) initiation [26]. As a country with the largest HIV epidemic

in the world [27], SA adopted the Joint United Nations Program on HIV/AIDS (UNAIDS) 90-90-90 testing and treatment targets with the objective of ending HIV/AIDS as a public health threat by 2030 [28, 29].

SA's national HIV testing guidelines are modelled on the WHO consolidated guidelines for HIV rapid testing to inform testing practices including quality assurance (QA) [30, 31]. In addition, the South African National Department of Health (SA NDoH) has a high quality management system that addresses all aspects of testing that exists in the country [30], and since 2016, the National Institute for Communicable Diseases (NICD), SA National Reference Laboratory for HIV testing, has been providing technical support for HTS in 27 PEPFAR-supported high HIV burden districts, under the HIV Rapid Test Quality Improvement Initiative (RTQII). The RTQII package involved: i) policy engagement ii) human resource iii) proficiency testing programs, iv) standardized registers and v) post-market surveillance. In SA, there is no national accreditation certification programme for HTS. The impact of the PEPFAR-funded RTCQI support was evaluated in HIV testing sites using two consecutive assessments [32]. ANC HIV testing sites were not included in this initiative. After the RTQII showed good results in many PEPFAR supported districts, the name changed from RTQII to Rapid Test Continuous Quality Improvement (RTCQI) and is being run by the SA NDoH. For simplicity, we will be referring to the initiative with the new name RTCQI. We set out to investigate HIV rapid testing practices within the PMTCT programme as the first step to assess whether PMTCT programme data can be used for surveillance. Our survey was limited to testing sites within ANC clinics, which were not part of the PEPFAR-funded RTCQI support. Following our survey, Woldesenbet et al. [33], conducted secondary analysis of routine HIV test results captured as part of ANSUR 2017, to assess the feasibility of using routine HIV data for surveillance. Their work did not assess HIV rapid testing practices.

## Methods

This paper is an analysis of the third objective (activity) of the broader study. The objectives of the study were (i) to assess the validity of routine versus survey data (planned to piggy-back onto the annual South African national antenatal survey), (ii) to review routine data quality and completeness abstracted from routine facility-based registers and (iii) to evaluate whether facilities adhered to the standardized protocols for rapid HIV testing (quality assurance of rapid HIV test results).

### Sampling strategy

Sampling was based on the second objective of the study (to ensure that adequate numbers of records were abstracted from facility-based ANC records). The sampling frame was developed from the list of facilities available in the 2014/15 SA NDoH District Health Information System dataset [34]. It comprised of the following: all public primary health care (PHC) facilities providing ANC (both ANSUR and non-antenatal survey sentinel sites (non-ANSUR) from all 9 provinces. These were classified according to locality type (rural and urban). Initially there were 36 strata that were defined from the sampling frame. Each stratum needed a facility contribution of at least 1% of the clinic data abstracted from facility-based records to provide enough data, hence smaller strata were merged, resulting in 26 strata in the final sampling frame and therefore these defined the main inclusion criteria. The mobile and referral clinics were excluded. Facilities sampled were randomly selected without replacement within each stratum. For the HIV rapid testing QA, the minimum feasible sample size allowing was determined at 95% confidence interval; with a margin of error of 5% was 360 facilities. The approach to sample size and sampling was designed to compare proportions of outcomes

between rural and urban localities, ANSUR and non-ANSUR participating sites and provinces, rather than being designed for external validity at sub-regional level.

## Data collection and instruments

Between February and May 2017, trained field workers conducted an audit of selected ANC sites offering routine HTS. We used the paper-based SPI-RT checklist for data collection which was adapted from the WHO SPI-RT checklist Version 3.0. to the local context. Data collection involved i) interviewing and observing one HIV tester (selected by the facility manager) in every testing site performing one simulated HIV rapid testing procedure and ii) auditing the site to assess compliance to the SA NDoH HTS Guideline [30]. The SPI-RT checklist had seven domains (totalling 64 points, with individual scores ranging from 5 to 12 points). The domains (scores in brackets) were: (i) training/certification (10), (ii) physical facility (5), (iii) safety (11), (iv) pre-testing phase (12), (v) testing phase (9), (vi) post-testing phase (9), and (vii) EQA (8) (Table 1). Each domain had quality indicator sub-elements rated 1, 0.5 or 0, where sites were 'compliant', 'partially-compliant' or 'non-compliant', respectively [35]. The SPI-RT checklist contains a built-in analysis function. The points were totaled and divided by the total number of possible points to produce an overall median score. Five levels were assigned in meeting the QA requirements ranging from level 0 to level 4 as follows:

- Level 0 site; a score of less than 40%, needs improvement in all areas and immediate remediation

- Level 1 site; a score between 40–59%, needs improvement in specific areas

- Level 2 site; a score between 60–79%, is partially ready for national site certification

- Level 3 site; a score between 80–89%, is close to national site certification

- Level 4 site; a score of 90% or higher–eligible for national site certification.

## Data analysis

Data were uploaded into an Open Data Kit software system (University of Washington, Washington DC., USA (https://opendatakit.org/), and exported to Excel (Microsoft Corporation, USA). STATA/SE14.0 (STATA Corporation, College Station, Texas, USA) was used for

**Table 1. Median scores and percentages by domain: Quality of HIV rapid testing practices, South Africa.**

| Domain | Median score (IQR)* | Median overall score* (IQR) as percentage of highest possible score |
|---|---|---|
| **Personnel training / certification** | 3.5 (1.0–5.0) | 35.0% (10.0%–50.0%) |
| **Physical facility** | 4.5 (4.0–5.0) | 90.0% (80.0%–100.0%) |
| **Safety** | 8.5 (7.0–10.0) | 77.3% (63.6%– 90.9%) |
| **Pre-testing phase** | 10 (9.0–11.0) | 83.3% (75.0% –91.7%) |
| **Testing phase** | 5.5 (3.0–7.0) | 61.1% (33.3% –77.8%) |
| **Post-testing phase** | 7 (5.5–8.0) | 77.8% (61.1% –88.9%) |
| **EQA** | 1 (0.0–4.0) | 12.5% (0.0%–50.0%) |
| **Overall score** | **39.8 (32.5–46.0)** | **62.1% (50.8%–71.9%)** |

** Note: the highest possible scores for each domain: personnel training /certification = 10; physical facility = 5; safety = 11; pre-testing phase = 12; testing phase = 9; post-testing phase = 9; and EQA = 8. IQR = Interquartile Range

analyses. We used descriptive statistics to calculate frequencies, medians and interquartile range (IQR); rank sum tests to compare differences; logistic regression to model associations and 95% confidence intervals (CI) to assess statistically significant differences between different site. We used descriptive statistics to calculate median percentage scores within a domain (domain-specific median score), and across all domains (overall median scores). The latter was used to classify testing sites according to five implementation levels for national site certification (see section above).

### Ethical considerations

The study protocol was approved by the South African Medical Research Council (SAMRC) Ethics Committee, (EC029-9/2015). "The protocol was also reviewed in accordance with the Centers for Disease Control and Prevention (CDC) human research protection procedures and was determined to be research, but CDC investigators did not interact with human subjects or have access to identifiable data or specimens for research purposes". All participants (facility manager and one designated HIV tester per site) provided written informed consent before participation.

## Results

### Facilities visited by province and sample size realization

We conducted QA in 346 of the selected 360 (96.1%) facilities across nine provinces (S1 Table). Twelve facilities were excluded because they were referral or mobile facilities and there were logistical challenges in two facilities. Sample size realization per province ranged between 87.9% and 100%. The majority (71.1%) of facilities assessed were in PEPFAR-supported districts and 63.9% were not antenatal sentinel sites (non-ANSUR).

### Median score and median percentage score by domain

Facilities obtained a median overall score of 39.8 (IQR 32.5–-46.0; total = 64), which corresponded to a median overall percentage score of 62.1% (IQR 50.8 –- 71.9%). The lowest median percentage scores were obtained under the domains training/certification (35.0% IQR 10.0–50.0%) and EQA (12.5% IQR 0.0–50.0%) (Table 1).

There were statistically significant inter-provincial differences in overall median percentage scores ranging from 46.9% (IQR 43.0 –- 59.4%) and 43.4% (IQR 37.5–48.0%), in Free State (FS) and Northern Cape (NC) provinces, respectively to 71.1% (IQR 67.2–78.1%) and 70.3% (IQR 63.3–78.1%) in Limpopo and Mpumalanga provinces (LP and MP), respectively (P value <0.001) (Table 2).

### Overall scores by geographical type, PEPFAR support, and participation in 2015 antenatal survey

Facilities in PEPFAR-supported priority districts had significantly higher median overall percentage scores (65.6% IQR 53.9–74.2%) compared to non–PEPFAR-supported facilities (56.6% IQR 47.7–66.0%) (Table 3).

### Implementation levels by province

Overall, most facilities (98.8%) did not meet the standard to qualify for national site certification in HIV rapid testing practices. Eighty-nine percent of sites were at level 2 and below with 37% (128/346 requiring improvement in specific areas (level 1) and 6.4% in all areas and immediate remediation (level 0). In three provinces (FS, NC and WC) none of the facilities

**Table 2. Distribution of median overall score and percentage by province: Quality of HIV rapid testing practices, South Africa.**

| Province | Planned sample size Number (%)* | Number (%) assessed** | Median overall scores (IQR)*** | Median overall score (IQR) as percentage of highest possible score* **† |
|---|---|---|---|---|
| Eastern Cape (EC) | 43 (11.9) | 43 (100.0) | 37.0 (31.0–45.0) | 57.8% (48.4–70.3%) |
| Free State (FS) | 17 (4.7) | 17 (100.0) | 30.0 (27.5–- 38.0) | 46.9% (43.0–59.4%) |
| Gauteng (GP) | 86 (23.9) | 80 (93.0) | 42.5 (36.3–47.3) | 66.4% (56.6–73.8%) |
| KwaZulu-Natal (KZN) | 78 (21.7) | 77 (98.7) | 38.0 (33.0–42.5) | 59.4% (51.6 –- 66.4%) |
| Limpopo (LP) | 45 (12.5) | 43 (95.6) | 45.5 (43.0–50.0) | 71.1% (67.2–78.1%) |
| Mpumalanga (MP) | 30 (8.3) | 29 (96.7) | 45.0 (40.5–50.0) | 70.3% (63.3–78.1%) |
| North West (NW) | 20 (5.6) | 20 (100) | 39.3 (34.8–52.3) | 61.3% (54.3–81.6%) |
| Northern Cape (NC) | 8 (2.2) | 8 (100) | 27.8 (24.0–30.75) | 43.4% (37.5–48.0%) |
| Western Cape (WC) | 33 (9.2) | 29 (87.9) | 33.0 (29.5–37.5) | 51.6% (46.1–58.6%) |
| **Total** | **360 (100)** | **346 (96.1)** | **39.8 (32.5–46.0)** | **62.1% (50.8–71.9%)** |

*As proportion of total number of facilities per province;

** As proportion of planned sample size;

***includes all seven domains; IQR = Interquartile Range

† test for equality of median percentage scores by province: p value <0.001.

were at implementation levels 3 and 4; and less than 2.6% of facilities in KwaZulu-Natal (KZN) were at implementation levels 3 and 4 (Table 4).

## Implementation levels by locality, site type and PEPFAR support

In a univariate logistic regression, facilities in PEPFAR-supported priority districts had 5.4 (95% CI: 1.6–17.8%) times higher odds of being at level 3 and 4 compared to non–PEPFAR–supported facilities (Table 5).

**Table 3. Score by locality, site type and PEPFAR support: Quality of HIV rapid testing practices, South Africa.**

| | Number (%) of facilities visited | Median score (IQR)* | Median Overall score (IQR) as percentage of highest possible score* | P. Value ** |
|---|---|---|---|---|
| **Locality** | | | | |
| Urban | 163 (47.1) | 39.5 (32.5 –- 45.0) | 61.7% (50.8 –- 70.3%) | 0.3 |
| Rural | 183 (52.9) | 40.0 (33.0–47.0) | 62.5% (51.6–73.4%) | |
| **Site type** | | | | |
| ANSUR facilities╪ | 125 (36.1) | 39.5 (32.0–46.0) | 61.7% (50–71.1%) | 0.5 |
| Non-ANSUR facilities | 221 (63.9) | 40.5 (33.0–46.5) | 63.3% (51.6–72.7%) | |
| **PEPFAR support** | | | | |
| Facilities in PEPFAR-supported districts | 246 (71.1) | 42.0 (35.0–47.5) | 65.6% (53.9–74.2%) | <0.001 |
| Non-PEPFAR facilities | 100 (28.9) | 36.3 (30.5–42.3) | 56.6% (47.7 –- 66.0%) | |
| **Overall** | **346 (100)** | **39.8 (32.5–46.0)** | **62.1% (50.8–71.9%)** | |

*includes all seven domains

** Wilcoxon rank sum tests; IQR = Interquartile Range

╪ ANSUR indicates that the facility participated in the 2015 antenatal survey

**Table 4. Distribution of implementation levels by province: Quality of HIV rapid testing practices, South Africa.**

| Province | Number of facilities | Level 0 (<40%) Number (%) | Level 1 (40–59%) Number (%) | Level 2 (60–79%) Number (%) | Level 3 (80–89%) Number (%) | Level 4 (90% orhigher) Number (%) |
|---|---|---|---|---|---|---|
| EC | 43 | 3 (7.0%) | 19 (44.2%) | 15 (34.9%) | 5 (11.6%) | 1 (2.3%) |
| FS | 17 | 3 (17.7%) | 11 (64.7%) | 3 (17.7%) | 0 (0%) | 0 (0%) |
| GP | 80 | 3 (3.8%) | 23 (28.8%) | 43 (53.8%) | 9 (11.3%) | 2 (2.5%) |
| KZN | 77 | 5 (6.5%) | 34 (44.2%) | 36 (46.8%) | 1 (1.3%) | 1 (1.3%) |
| LP | 43 | 0 (0%) | 2 (4.7%) | 33 (76.7%) | 8 (18.6%) | 0 (0%) |
| MP | 29 | 0 (0%) | 7 (24.1%) | 16 (55.2%) | 6 (20.7%) | 0 (0%) |
| NW | 20 | 1 (5%) | 8 (40.0%) | 6 (30.0%) | 5 (25.0%) | 0 (0%) |
| NC | 8 | 3 (37.5%) | 5 (62.5%) | 0 (0%) | 0 (0%) | 0 (0%) |
| WC | 29 | 4 (13.8%) | 19 (65.5%) | 6 (20.7%) | 0 (0%) | 0 (0%) |
| All | 346 | 22 (6.4%) | 128 (37.0%) | 158 (45.7%) | 34 (9.8%) | 4 (1.2%) |

Level 0 site [RED]; a score of less than 40, needs improvement in all areas and immediate remediation

Level 1 site [ORANGE]; a score between 40–59%, needs improvement in specific areas

Level 2 site [YELLOW]; a score between 60–79%, is partially ready for national site certification

Level 3 site [LIGHT GREEN]; a score between 80 –- 89%, is close to national site certification

Level 4 site [DARK GREEN]; a score of 90% or higher, is eligible for national site certification

## Commonly identified gaps

Table 6 lists performance gaps for each domain where testing sites were rated 0, including the number of facilities that were fully non-compliant for the gaps identified. These included 87% of sites not having documents indicating testers' competency prior to HIV testing; quality control (QC) specimens not used routinely and lack of corrective action for unsatisfactory PT results (Table 6).

## Discussion

We sought to investigate quality of HIV rapid testing practices in selected antenatal public health facilities across SA using an adapted SPI-RT checklist classification for national site

**Table 5. Stratified analysis of implementation levels by locality, site type and PEPFAR support: Quality of HIV rapid testing practices, South Africa.**

| Province | Number of facilities | Level 0 Number (%) | Level 1 Number (%) | Level 2 Number (%) | Level 3 Number (%) | Level 4 Number (%) | Odds ratio (95% CI) * |
|---|---|---|---|---|---|---|---|
| **Locality** | | | | | | | |
| Urban | 163 | 13 (8.0%) | 58 (35.6%) | 75 (46.0%) | 15 (9.2%) | 2 (1.2%) | 1.1 (0.6–2.2) |
| Rural | 183 | 9 (4.9%) | 70 (38.3%) | 83 (45.4%) | 19 (10.4%) | 2 (1.1%) | |
| **Site type** | | | | | | | |
| ANSUR | 125 | 7 (5.6%) | 50 (40.0%) | 58 (46.4%) | 8 (6.4%) | 2 (1.6%) | 0.6 (0.3–1.3) |
| Non-ANSUR | 221 | 15 (6.8%) | 78 (35.3%) | 100 (45.3%) | 26 (11.8%) | 2 (0.9%) | |
| **Support** | | | | | | | |
| PEPFAR support | 246 | 14 (5.7%) | 75 (30.5%) | 122 (49.6%) | 32 (13.0%) | 3 (1.2%) | 5.4 (1.6–17.8) |
| Non-PEPFAR support | 100 | 8 (8.0%) | 53 (53.0%) | 36 (36.0%) | 2 (2.0%) | 1 (1.0%) | |
| All | 346 | 22 (6.4%) | 128 (37.0%) | 158 (45.7%) | 34 (9.8%) | 4 (1.2%) | |

*univariate logistic regression assessed the odds of being at levels 3 and above (vs being at levels < 3) for facilities in PEPFAR districts, ANSUR facilities and rural facilities. CI- Confidence Interval

**Table 6. Commonly identified deficiencies: Quality of HIV rapid testing practices, South Africa.**

| Domains | Common gaps (% of facilities) |
|---|---|
| **Personnel training/ certification:** | Testers not trained on EQA/ PT and QC (56.1%) |
| | No evidence of refresher training/ no documentation (77.2%) |
| | No documents indicating testers' competency prior to HIV testing (87.0%). |
| | No national certification programme in place (84.4%) |
| **Physical facility** | Room temperature poorly monitored /test kits exposed to direct sunlight/ no thermometers (19.9%) |
| | No designated area for testing/room used for multiple purposes (7.8%) |
| **Safety** | Standard Operating Procedures (SOPs)/Job aids not available in the testing room (35%) |
| | Incorrect use of gloves /absence of protective aprons (8.1%) |
| **Pre-testing phase** | Job aids on client sample collection not available/posted (26.9%) |
| | Test kits not initialed, not dated (62.4%) |
| **Testing phase** | Timers not in good working order/not available (39.9%) |
| | Procedure followed inaccurately (14.2%) |
| | Incorrect amount of buffer used (8.7%) |
| | QC specimens not used routinely/no samples available (43.5%) |
| | Unsure how to handle invalid QC results/ not recorded (70.2%) |
| **Post-Testing phase** | Incomplete recording of QC key elements (lot number, expiry date) on HIV testing logbooks/registers (8.4%) |
| | Inaccurate/inconsistent capturing of total summaries in logbooks (19.9%) |
| | Invalid test results not recorded in register/ logbook (55.8%) |
| | Registers not properly labelled and archived when full (24.3%) |
| **EQA (PT, supervision & retesting)** | Testing point not enrolled in EQA/PT (62.7%) |
| | QC results are not reviewed by the person in charge (69.9%) |
| | Corrective action for unsatisfactory results implemented (86.1%) |
| | QC samples for PT not available/ PT not done (46.0%) |
| | Periodic supervisory visits not done/ not documented (72.3%) |
| | No feedback/ re-training (80.4%) |

certification. Our main finding demonstrates that in 2017, almost all (98.8%) ANC testing sites did not adequately meet HIV rapid testing standards for national site certification. Compared to results of the 2015/17 RTCQI consecutive national site assessments for HIV rapid testing QA measured at SA HTS sites, our results show that ANC clinics are at lower implementation levels compared to HTS sites. In the RTCQI programme conducted in HTS sites around at the same time as this study, 38.8% of HTS sites were eligible for national certification as compared to 11% of antenatal sites that were eligible for site certification in our study. The difference in site eligibility between our study and the RTCQI sites could be attributed to the implementation of the PEPFAR-funded RTCQI support in HTS sites which did not include ANC clinics.

According to the WHO, a score of at least 80% in each of the three phases (pre-testing, testing and post-testing) critical for the accuracy of diagnosis may be acceptable [3]. Our survey results demonstrate that the overall median percentage scores for both testing and post-testing phases were below 80%. This finding has important implications given the increase in the number of clients requiring HTS [36, 37] and high PMTCT uptake of 99.8% in SA; because high quality performance is required on all 3 phases. Consistent with our findings, it is encouraging to note that overall median scores for pre-testing in our study (i.e. 83.3%) were similar to those reported during the RTCQI programme assessments [32]. The RTQI assessment and

our survey differed in the testing phase median scores; with our study scoring 16.7 percentage points below that of the PEPFAR-funded RTCQI programme [32].

Our study demonstrates that sites scored poorly on training/certification and EQA/PT domains. Similarly, in the PEPFAR-funded RTCQI programme [32] the median percentage scores for training and certification in HTS site assessments remained the lowest in both consecutive assessments (30.0% (IQR 20.0–40.0%) and 50.0% (IQR: 40.0–60.0%) in the first and second assessments, respectively) [32]. This lack of improvement in training and certification in the PEPFAR-funded RTCQI assessments could be attributed to the lack of national certification programme and the fact that sometimes the training may not be properly cascaded down to staff in the respective clinics [32]. Stepping up staff competencies is critically important given clients' rights and the evidence suggesting that pregnant women value and appreciate when screening procedures are conducted by knowledgeable, supportive and respectful health care providers as this allows for a positive pregnancy experience [31, 38, 39].

We identified two common domain-specific gaps in our study where a high proportion of testing sites were rated fully non-compliant. These include non-existence of a nationally accredited programme and non-adherence to the EQA/PT programme. Although there is a system for assessing user competency in SA there is no nationally accredited certification programme, and the system does not work well. The HTS policy recommends quarterly EQA/PT supervisory support visits to testing sites [30]. It is apparent that current HIV RT training and supervisory mechanisms are weak and need to be strengthened and monitored. Additionally, we found that some QC assay procedures were not adhered to. These included incorrect use of buffer drops; exposing test kits to sunlight and incorrect time to reading results. Engel et al. [40] found that frequently changing HIV test kits brands were important barriers contributing to the use of incorrect amounts of buffer drops and shortened time to reading results, respectively [41]. We found that there was non-compliance to HIV testing standards and poor documentation which are suggestive of low staff competency levels. These findings are consistent with previous studies reporting high prevalence of HIV false-positive and false-negative rapid testing results [8, 9, 21, 41] and poor data quality [6, 42–45]. In 2019, Woldesenbet et al. [33], using data from the 2017 ANSUR, reported high PPA (97.6%) between third generation point of care (POC) rapid testing and laboratory-based fourth generation immunoassay test. Despite the low quality of testing practice we report in this study, it is encouraging that the risk of being diagnosed as false HIV positive was minimal in the above study [33]. However, since the study (Woldesenbet et al) was based on a cross-sectional survey, it will be important to monitor test agreement on a regular basis (by using other methods such as PT), and caution needs to be taken in using PMTCT data for monitoring of antenatal HIV prevalence [33].

Our further analyses show that ANC testing sites in PEPFAR-supported districts were more likely of being at level 3 and 4 compared to those in non-PEPFAR-supported districts. This difference could be due to the additional technical and resource support provided by PEPFAR to these facilities [32, 46]. A spill-over effect could have been possible within the same facility from HTS testers who receive PEPFAR-funded RTCQI support. It is common for HIV testers to rotate between HTS and ANC services within the same facility to address human resource shortages [47]. Between 2015 and 2016, an audit of rural antenatal PHC clinics in KZN reported an average rating score of 64.4% (CI:44% ± 84%) and 89.2% (CI: 74% ± 100%), respectively in terms of compliance to the WHO guidelines for assuring accuracy and reliability of HIV rapid tests [7]. The PEPFAR-funded RTCQI programme also reported variations in implementation levels across provinces [32].

Our study had some reassuring findings. Privacy is an important determinant to HIV testing acceptance. Our findings show that facilities scored highest (90%) in the physical facility domain. In most sites, national HIV testing guidelines were available and good stock

management systems were in place. Consistent with our finding, evaluations by Jones et al. [48] in 2019 on implementation of PMTCT policies (2013–2016) found that the highest proportions (around 60%) of facilities in rural SA reported no stock-outs of HIV test kits in the year before the survey [48]. However, contrary findings were reported by two studies in SA; a qualitative study [41] conducted in Durban, Cape Town and EC (2012–2013) and a survey [24] in rural PHC facilities in KZN (2015–2016). These studies identified HIV test kits stockout as a major barrier which resulted in poor compliance with testing guidelines, with some testers having to adapt HIV rapid testing algorithms [24, 41] or to refer clients to nearby facilities [41]. Focused continuous quality improvement (CQI) interventions including technical on-site mentoring have shown to remarkably improve PMTCT data quality [42, 43].

## Study strengths and limitations

The strength of our study is that ANC facilities were selected from a range of national scenarios, considering locality, site type and PEPFAR support. Furthermore, field workers were carefully trained and supervised. In addition, data collection occurred outside the period (October) when the regular ANSUR survey took place to minimize bias.

Our study had several limitations. Firstly, only one tester was observed on one occasion from each site performing a simulated HIV test; so, it is possible that testers assessed were not competent in HIV rapid testing, and nuances of observing actual testing in real life may have been missed during simulation. Secondly, smaller volume, referral and mobile testing sites were not included. Thirdly, we did not set up the sample size to provide provincially or nationally representative findings.

## Conclusions

Our findings add to the growing literature on the utility of routine PMTCT data for monitoring antenatal HIV prevalence. It suggests that the growing demand for HIV rapid testing in the era of UTT has not kept pace with QA standards within the PMTCT programme, since almost all antenatal sites did not meet the criteria for national site certification. Substantial progress has been reported in HTS sites following the implementation of the PEPFAR-funded RTCQI support in SA. Despite implementation gaps observed in our study, many focused CQI strategies have been introduced by the SA NDoH as steps towards preparing for routine PMTCT data-based surveillance. These include improving the 2017 ANSUR sampling strategy, the Ideal Clinic Realization and Maintenance Programme (a comprehensive approach aiming at creating an enabling environment for sustainable implementation of QA processes), rationalization of registers (aimed at reducing the number of registers used in PHC facilities), and Nurse—connect (a project that uses mobile technology to send targeted support messages and expert information to nurses on maternal and child health) [49–51]. Improving HIV rapid testing practices in ANC clinics would be invaluable to speed up South Africa's readiness to transition to routine programme data—based surveillance. We recommend expanding PEPFAR-funded RTCQI support, using the WHO SPI-RT tool to all HIV testing sites, including ANC sites. In future a non-simulated nationally representative survey should be conducted to evaluate the overall quality of HIV rapid testing practices in PHC settings in South Africa.

## Supporting information

**S1 Table. Distribution of facilities according to province, sentinel site and locality type in South Africa.**
(DOCX)

## Acknowledgments

We would like to acknowledge our funders, collaborators, the management and staff from the SA NDoH, Epicentre, SAMRC project team and research participants.

## Author Contributions

**Conceptualization:** Ameena Goga.

**Formal analysis:** Selamawit A. Woldesenbet, Adrian Puren, Vincent I. Maduna, Carl Lombard.

**Funding acquisition:** Ameena Goga.

**Investigation:** Duduzile F. Nsibande, Selamawit A. Woldesenbet, Adrian Puren, Peter Barron, Vincent I. Maduna, Carl Lombard, Mireille Cheyip, Mary Mogashoa, Yogan Pillay, Vuyolwethu Magasana, Trisha Ramraj, Tendesayi Kufa, Ameena Goga, Witness Chirinda.

**Methodology:** Duduzile F. Nsibande, Adrian Puren, Carl Lombard, Mary Mogashoa, Vuyolwethu Magasana, Trisha Ramraj, Tendesayi Kufa, Ameena Goga.

**Project administration:** Duduzile F. Nsibande.

**Supervision:** Ameena Goga, Witness Chirinda.

**Writing – original draft:** Duduzile F. Nsibande.

**Writing – review & editing:** Selamawit A. Woldesenbet, Adrian Puren, Peter Barron, Vincent I. Maduna, Carl Lombard, Mireille Cheyip, Mary Mogashoa, Yogan Pillay, Vuyolwethu Magasana, Trisha Ramraj, Tendesayi Kufa, Gurpreet Kindra, Ameena Goga, Witness Chirinda.

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
