## [Decision Letter · Decision Letter 0]

3 Mar 2022

PONE-D-22-00472Investigating the quality of HIV rapid testing practices in public antenatal health care facilities, South AfricaPLOS ONE

Dear Dr. Nsibande,

Thank you for submitting your manuscript to PLOS ONE. After careful consideration, we feel that it has merit but does not fully meet PLOS ONE’s publication criteria as it currently stands. Therefore, we invite you to submit a revised version of the manuscript that addresses the points raised during the review process.

ACADEMIC EDITOR:Mansucript submitted inspects/validates quality of  HIV rapid testing practices in public antenatal health care facilities. Authors had  made an attempt to put forth the gaps in practices through application of valid tool WHO Stepwise-Process-for-Improving-the-Quality-of-HIV-Rapid-Testing (SPI-RT) checklist. Please modify/accept more concise introduction as suggested by reveiwer 1. Sampling section needs to be elaborated  further as suggested by reveiwer 1. Conculsion section may further be added with recommendations as marked by revewier 2 apart from other queries raised. Specific feedbackGap identification and gap closure is an important tool for health system strengthening. Application of WHO tool to addresss that ,would strengthen the systems in place.  Please submit your revised manuscript by Apr 17 2022 11:59PM. If you will need more time than this to complete your revisions, please reply to this message or contact the journal office at plosone@plos.org. Please include the following items when submitting your revised manuscript:A rebuttal letter that responds to each point raised by the academic editor and reviewer(s). You should upload this letter as a separate file labeled 'Response to Reviewers'.A marked-up copy of your manuscript that highlights changes made to the original version. You should upload this as a separate file labeled 'Revised Manuscript with Track Changes'.An unmarked version of your revised paper without tracked changes. You should upload this as a separate file labeled 'Manuscript'.

We look forward to receiving your revised manuscript.

Kind regards,

Gopal Ashish Sharma, MBBS, MD

Academic Editor

PLOS ONE

Journal Requirements:

“The funders had no role in study design, data collection and analysis decision to publish, or preparation of the manuscript.”

Reviewers' comments:

Reviewer's Responses to Questions

**Comments to the Author**

1. Is the manuscript technically sound, and do the data support the conclusions?

Reviewer #1: Yes

Reviewer #2: Yes

2. Has the statistical analysis been performed appropriately and rigorously? 

Reviewer #1: Yes

Reviewer #2: Yes

3. Have the authors made all data underlying the findings in their manuscript fully available?

Reviewer #1: No

Reviewer #2: Yes

4. Is the manuscript presented in an intelligible fashion and written in standard English?

Reviewer #1: Yes

Reviewer #2: Yes

5. Review Comments to the Author

Reviewer #1: The Introduction part can be reduced, it is bit lengthy. Although the matter of introduction is relevant, but it can be concised.

The Sampling strategy may be elaborated: what was sampling unit, how the sample size calculated, how strata's were defined and selected and finally how were the facilities selected.

Inclusion and exclusion criteria may be included.

Reviewer #2: 1. The study was conducted in 2017. Are there any changes in National HIV strategy since then? Are the findings relevant in 2022?

2. Elaborate RTCQI in Abstract section as it is not a common abbreviation (Line 24).

3. Is it “Countries” or “Districts”? (Line 93)

4. Statistical difference (P value) among the Provinces can be stated for better understanding. (Table 2)

5. Conclusion section should contain detailed recommendations on the basis of study observations. Moreover, based on study limitations, suggestions can be made for future research studies for generating more generalizable and valid findings.

6. PLOS authors have the option to publish the peer review history of their article (what does this mean?). If published, this will include your full peer review and any attached files.

Reviewer #1: No

Reviewer #2: No

---

## [Author Response · Author response to Decision Letter 0]

13 Apr 2022

Reviewer’s Comments PONE-D-22-00472

Specific feedback:

Reviewer #1: 

1.The Introduction part can be reduced, it is bit lengthy. Although the matter of introduction is relevant, but it can be concised.

Response: The introduction has been reduced as suggested from 1 050 to 851 words (pages 5-8).

2. The Sampling strategy may be elaborated: what was sampling unit, how the sample size calculated, how strata's were defined and selected and finally how were the facilities selected.

Inclusion and exclusion criteria may be included.

Response: The sampling strategy has been revised under methods page 9 (lines 114-140)

Reviewer # 2

1. The study was conducted in 2017. Are there any changes in National HIV strategy since then? Are the findings relevant in 2022?

Response: Response: The findings are still relevant in 2022. South Africa is still implementing the 2017-2022 National HIV Strategic Plan aims at an intensified focus on districts and locations with high burdens of HIV, STIs and/or TB; on adolescent girls and young women and on tailoring interventions for the key and vulnerable populations. 

2. Elaborate RTCQI in Abstract section as it is not a common abbreviation (Line 24).

Response: The full name for RTCQI is ‘Rapid Test Continuous Quality Improvement. It has been added in the abstract on page 5 (line 24). 

3. Is it “Countries” or “Districts”? (Line 93). 

Response: The word countries has been replaced with districts under Introduction, on page 8 (line 96)

4. Statistical difference (P value) among the Provinces can be stated for better understanding. (Table 2)

Response: statistical significance test (p-value) has been added under results, page 13 Table 2 (lines 205-6)

5. Conclusion section should contain detailed recommendations on the basis of study observations. Moreover, based on study limitations, suggestions can be made for future research studies for generating more generalizable and valid findings.

Response: The conclusion section has been revised. Detailed recommendations based on study limitations have been made on pages 26 (lines 365-378)

---

## [Decision Letter · Decision Letter 1]

6 May 2022

Investigating the quality of HIV rapid testing practices in public antenatal health care facilities, South Africa

PONE-D-22-00472R1

Dear Dr. Duduzile Faith Nsibande,

We’re pleased to inform you that your manuscript has been judged scientifically suitable for publication and will be formally accepted for publication once it meets all outstanding technical requirements.

Kind regards,

Gopal Ashish Sharma, MBBS, MD

Academic Editor

PLOS ONE

Additional Editor Comments (optional):

Editor -Specific 

The manuscript submitted  and further reviewed addresses gaps and quality of HIV rapid testing practices in public antenatal health care facilities by use of valid tool by WHO. 

The manuscript highights key vital concern of Quality , as only 11% of assessed health care  facilities had score close to 80% and above (level 3 & 4). As the process itself is rapid and quintessential to primarily manage ANC patients , the quality concerns warrant sequential and regular evaluvation to curtail HIV transmission at grassroot level. Identified gaps,specifically in the documentation processes along with national certfication /accrediation  etc as discussed in the manuscript needs priortized intervention by policy makers. This would further contribute in HSS across ANC services in the country. 

Reviewers' comments:

Reviewer's Responses to Questions

**Comments to the Author**

1. If the authors have adequately addressed your comments raised in a previous round of review and you feel that this manuscript is now acceptable for publication, you may indicate that here to bypass the “Comments to the Author” section, enter your conflict of interest statement in the “Confidential to Editor” section, and submit your "Accept" recommendation.

Reviewer #1: All comments have been addressed

Reviewer #2: All comments have been addressed

2. Is the manuscript technically sound, and do the data support the conclusions?

Reviewer #1: Yes

Reviewer #2: Yes

3. Has the statistical analysis been performed appropriately and rigorously? 

Reviewer #1: Yes

Reviewer #2: Yes

4. Have the authors made all data underlying the findings in their manuscript fully available?

Reviewer #1: Yes

Reviewer #2: Yes

5. Is the manuscript presented in an intelligible fashion and written in standard English?

Reviewer #1: Yes

Reviewer #2: Yes

6. Review Comments to the Author

Reviewer #1: The suggestions as made, have been taken into consideration by the author and have been addressed too.

Reviewer #2: (No Response)

7. PLOS authors have the option to publish the peer review history of their article (what does this mean?). If published, this will include your full peer review and any attached files.

Reviewer #1: No

Reviewer #2: No

---

## [Editor Report · Acceptance letter]

19 Aug 2022

PONE-D-22-00472R1 

Investigating the quality of HIV rapid testing practices in public antenatal health care facilities, South Africa     

Dear Dr. Nsibande:

I'm pleased to inform you that your manuscript has been deemed suitable for publication in PLOS ONE. Congratulations! Your manuscript is now with our production department. 

Kind regards, 

on behalf of

Dr. Gopal Ashish Sharma 

Academic Editor

PLOS ONE